# Genomic analysis on pygmy hog reveals extensive interbreeding during wild boar expansion

Langqing Liu [1], Mirte Bosse [1], Hendrik-Jan Megens[1], Laurent A.F. Frantz[2,3], Young-Lim Lee[1], Evan K. Irving-Pease [3], Goutam Narayan [4,5], Martien A.M. Groenen [1] & Ole Madsen[1]

Wild boar (*Sus scrofa*) drastically colonized mainland Eurasia and North Africa, most likely from East Asia during the Plio-Pleistocene (2–1Mya). In recent studies, based on genome-wide information, it was hypothesized that wild boar did not replace the species it encountered, but instead exchanged genetic materials with them through admixture. The highly endangered pygmy hog (*Porcula salvania*) is the only suid species in mainland Eurasia known to have outlived this expansion, and therefore provides a unique opportunity to test this hybridization hypothesis. Analyses of pygmy hog genomes indicate that despite large phylogenetic divergence (~2 My), wild boar and pygmy hog did indeed interbreed as the former expanded across Eurasia. In addition, we also assess the taxonomic placement of the donor of another introgression, pertaining to a now-extinct species with a deep phylogenetic placement in the *Suidae* tree. Altogether, our analyses indicate that the rapid spread of wild boar was facilitated by inter-specific/inter-generic admixtures.

[1] Animal Breeding and Genomics, Wageningen University & Research, 6708PB Wageningen, the Netherlands. [2] School of Biological and Chemical Sciences, Queen Mary University of London, E1 4NS, London, United Kingdom. [3] Palaeogenomics and Bioarcheology Research Network, Research Laboratory for Archeology and History of Art, University of Oxford, Oxford OX1 3QY, United Kingdom. [4] Durrell Wildlife Conservation Trust, Les Augrès Manor, Jersey JE3 5BP Channel Islands, United Kingdom. [5] Pygmy Hog Conservation Programme, EcoSystems-India, Indira Nagar, Basistha, Guwahati, Assam 781029, India. Correspondence and requests for materials should be addressed to L.L. (email: langqing.liu@wur.nl) or to O.M. (email: ole.maden@wur.nl)

The expansion of species into novel habitats can have tremendous impacts on the native fauna. If the native fauna contains species that are closely related to the invasive population, hybridization may also threaten the integrity and survival of native species[1]. Observation of admixture in expanding populations has led to speculation whether admixture has an important role in driving the success of those populations[2–4]. Although the old-world pigs, the *Suidae*, are distributed throughout Africa and Eurasia, only two species are recognized across mainland Eurasia: wild boar (*Sus scrofa*) and pygmy hog (*Porcula salvania*)[5–9].

This, however, was not always the case and extensive fossils records suggest that Eurasia hosted a highly diverse set of *Suidae* species that originated during the Miocene[10–13] (Fig. 1a). In middle Miocene, the first isolated lineage to split from those early *Suidae* was *Babyrousa*, which forms an ancient lineage endemic to the island of Sulawesi (Fig. 1b)[14,15]. Along with the global cooling during the late Miocene[16], a new subfamily, the *Suinae*, emerged in the fossil record[17,18] and replaced almost all other subfamilies of *Suidae* present at that time[19,20] (Fig. 1c). The *Suinae* later diversified into multiple tribes[17,18]. This was followed by a divergence of the African *Suidae* and the Eurasian *Sus* genus, at around the Miocene/Pliocene boundary (Fig. 1d). Shortly thereafter, the divergence within the *Sus* genus started during the early Pliocene (Fig. 1f). Several *Sus* species on the islands of southeast Asia (ISEA) evolved during the early/mid Pliocene[21]. Relatively high levels of species diversity were likely maintained across Eurasia, until the early Pleistocene, when wild boar expanded out of East Asia into almost every type of ecosystem across the old-world. This expansion was highly efficient mirroring the great human expansion during the late Pleistocene[22], and previous study have suggested that it is the reason for the disappearance of most suid species across Eurasia[23,24] (Fig. 1g, h). With this, the layout for the modern *Suidae* species became settled.

As the only reminiscence of the once highly diverse Pliocene suid fauna, the pygmy hog represents not only an important species to conserve but also the key to our understanding of the expansion of wild boar. Indeed, previous studies have suggested that the rapid and highly efficient expansion of wild boar was facilitated by interspecific adaptive gene-flow[21,25]. Under this scenario, wild boar absorbed rather than replaced species, a process paralleled to some extent in humans[26–28]. Although pygmy hog is now highly endangered and restricted in a small corridor of high grassland at the southern foothills of the Himalaya[29–31], it was far more widespread in the past[32] (Fig. 1e). According to middle Pleistocene fossil remains from southwest China, the geographical range of pygmy hog and wild boar did

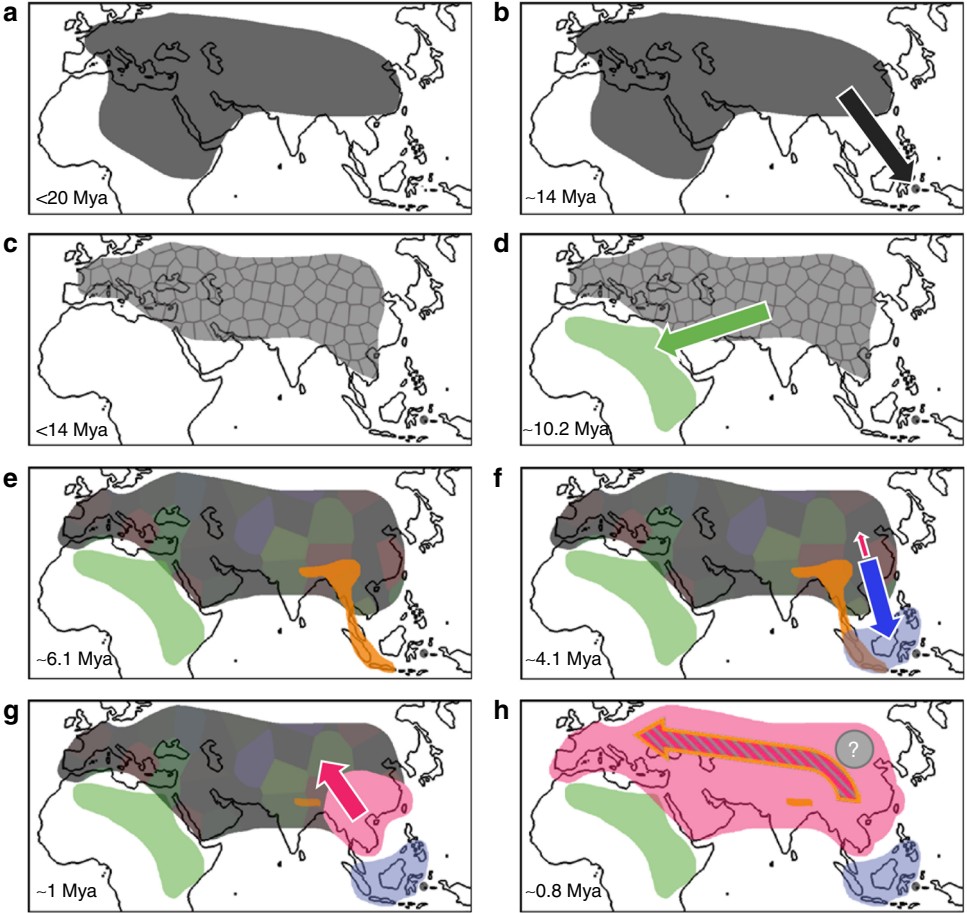

**Fig. 1** A series of schematic models depicting the geographic evolution of *Suidae* species over the past 20 Mya. **a** Emergence of Suidae across Eurasian and Northern Africa. **b** The black arrow depicts the hypothesized trajectory of *Babyrussa* migrating to ISEA. **c** Emergence of *Suinae* (gray collage pattern) that replaced other *Suidae*. **d** The green arrow indicates the diversification of the ancestor of Sub-Saharan suids. **e** Eurasian *Suinae* split into multiple genera (multi-color collage pattern), including pygmy hog (orange shade). **f** The blue arrow depicts the migrations of *Sus* to ISEA. The red arrow indicates emergence of *Sus scrofa*. **g, h**. The spread of *Sus scrofa* from southern Asia to Europe and replacement of all Suinae species except pygmy hog. During the replacement, Sus scrofa populations introgressed at least three times with one of these Suinae species (ghost linage), pygmy hog and ISEA *Sus*, respectively. The colors correspond to those used in Fig. 2 and represent the cluster on the tree to which the samples belong

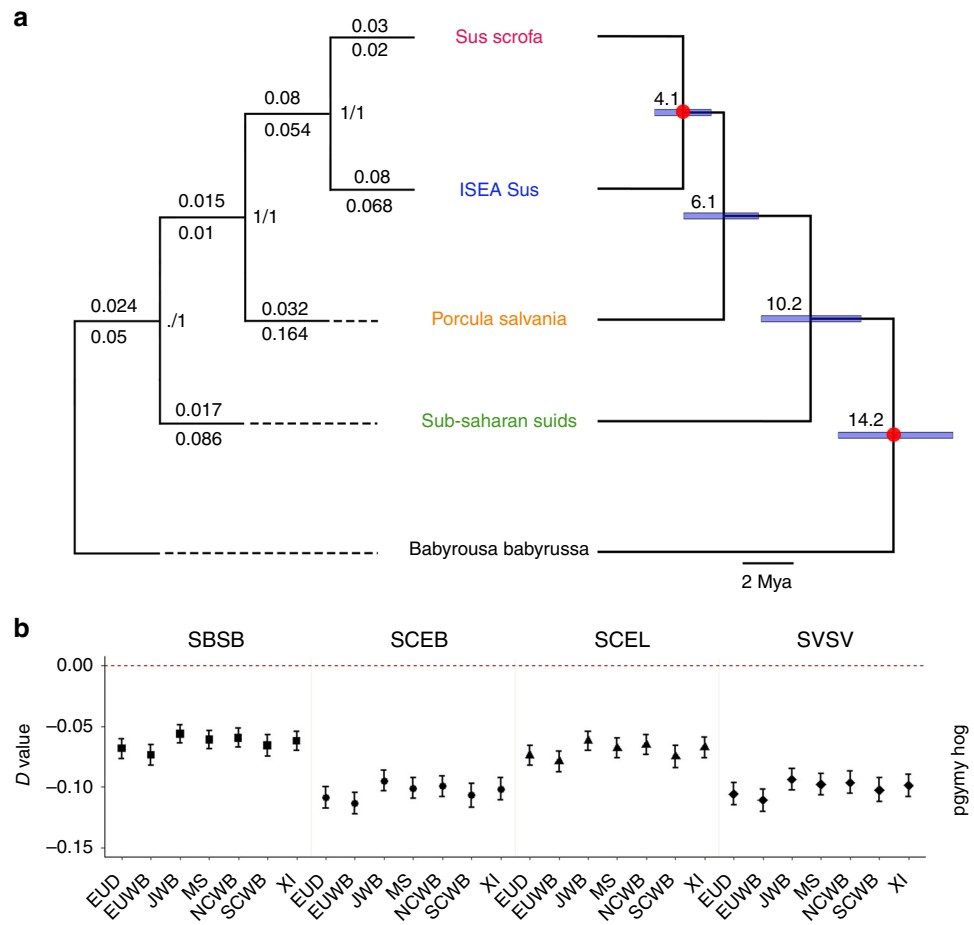

**Fig. 2** Phylogenetic relationships and divergence of the *Suidae* species used in the current study and admixture event between pygmy hog and *Sus scrofa*. **a** The tree on the left is the ML tree of the Suidae family based on consensus and concatenation methods. Branch length of the consensus analysis above branches, concatenation below branches. Node labels show bootstrap values of the concatenation analysis and concordance factors of the consensus analysis, respectively. The tree on the right is the time tree of divergence. Node labels show age in million years. Blue bars indicate 95% confidence interval and red dots show the calibration points (See Supplementary material Figs. 1–3 for full trees). **b** A diagram depicting the excess derived allele sharing when comparing sister taxa and outgroups. Each column contains the fraction of excess allele sharing by a taxon (up/down) with the pygmy hog/outgroup compared with its sister taxon (up/down). We computed D-statistics of the form D (X, Y, Pygmy hog, warthog). Error bars correspond to three standard errors. (SBSB = *Sus barbatus*, SCEB = *Sus cebrifons*, SCEL = *Sus celebensis*, SVSV = *Sus verrucosus*, EUD = European domesticated pig, EUWB = European wild boar, JWB = Japanese wild boar, MS = Meishan, NCWB = Northern China wild boar, SCWB = Southern China wild boar, XI = Xiang)

overlap[33,34], implying that the temporal and geographical proximity of pygmy hog and wild boar could have resulted in hybridization. Therefore, pygmy hog provides an excellent comparative framework to study the evolutionary processes that occurred during a fast and extensive radiation. We analyzed six genomes of pygmy hog in combination with genomes of related suid species and found strong support for an important role of inter-species hybridization during range expansion. The results suggest that wild boar hybridized with pygmy hog and a now-extinct suid species during the rapid spread across Eurasia and North Africa.

## Results

**Phylogenetic relationships and divergence of the *Suidae* species.** We sequenced the genomes of six pygmy hogs and one *Babyrousa babyrussa* and analyzed these with the genome sequences of 31 individuals, in total, representing 10 of the extant *Suidae* species (See Material and methods and Supplementary data 1 for details). We first assessed the phylogenetic relationship of these species. The concatenation and consensus methods resulted in the same main topology (Fig. 2a, Supplementary

Figs. 1, 2). The phylogenetics analyses clearly show that the most basal split within the *Suinae* are sub-Saharan suids followed by a highly supported split of pygmy hogs (BS = 100 in supermatrix and CF = 1 in supertree) from all *Sus* species. To compare our phylogenetics results to an earlier study using fragments of mitochondrial DNA[35], we also carried out a Bayesian phylogenetic analysis using complete mitochondrial genomes (Supplementary Fig. 3). The resulting topology is consistent with previous studies confirming pygmy hog as basal to *Sus*. Thus, both the genome-wide autosomal phylogenetic analysis and complete mitochondrial genome analysis support, with very high confidence, that pygmy hog is a monophyletic sister taxon of *Sus*.

We selected autosomal genomic loci supporting the main topology (Fig. 2a, Supplementary Fig. 4). Molecular clock analyses indicated that the divergence between Sub-Saharan suids and Eurasian *Suidae* (pygmy hog and *Sus*) took place shortly after the divergence of *Babyrousa babyrussa* ~ 10.2 Mya (95% highest posterior density (HPD) = 12.7–7.9). Pygmy hog separated from the common ancestor with *Sus* during the early Pliocene, ~6.1 Mya (95% HPD = 7.8–4.2) and the Eurasian wild boar split from

other *Sus* species during the early Pliocene ~ 4.1 Mya (95% HPD = 5.5–2.7). (Fig. 2a, Supplementary Fig. 4, Supplementary Note)

**Admixture between pygmy hog and wild boar**. In order to test whether temporal and geographical proximity of closely related species could have resulted in hybridization, we looked for interspecific admixture signal within our genome-wide data. Several studies have reported that in diverged species sex-linked markers may show evolutionary histories incongruent to other sex-linked and/or autosomal markers[36–40]. Thus, we analyzed autosomes and the X chromosome separately. To investigate whether any of the sequenced pygmy hogs showed evidence for autosomal introgression from ancestors of present-day *Sus* species, or vice-versa, the pygmy hog was compared with representatives of eleven *Sus* populations using *D*-statistics. We found a significant overrepresentation of derived alleles between the pygmy hog and mainland Eurasian wild boar at autosomal chromosomes (Fig. 2b, Supplementary data 2), indicative of admixture. This signal of admixture was also supported by TreeMix analysis. The best-fitting model suggests an ancient admixture between ancestral wild boar and pygmy hog (Supplementary Fig. 5).

To further examine this autosomal genome-wide pattern of admixture between pygmy hog and wild boar, we combined *D*-statistics and fd to infer regions of introgression. We also calculated DNA sequence divergence ($d_{xy}$) for each window to reduce false-positive signals[41]. This resulted in 636 putative introgression intervals between pygmy hog and wild boar from Europe, North China and South China, of which 427 (67.1%) are shared within wild boars (Supplementary Fig. 6, Supplementary Note). This suggests the admixture occurred before the divergence within Eurasian wild boar, further sustaining the evidence of an ancestral gene-flow between pygmy hog and the common ancestor of wild boar.

**Wild boar harbors genetic introgression from a ghost linage**. We next addressed evidence of admixture from the X chromosome starting with reconstructing a ML tree for the X chromosome. This tree displays a different topology compared with our main phylogenetic tree (Supplementary Fig. 8). Previous study has reported two distinct haplotypes in European pigs and South Asian pigs, and proposed that this might be derived from a now-extinct Suid (ghost) lineage[40]. To investigate the existence of genealogical discordance, we carried out a sliding-window *D*-statistic and a machine-learning based detection of local phylogenetic incongruent regions[42]. Both approaches supported that there is a ~ 40.6 Mbp (46.2–86.8 Mbp) region on the X chromosome, where pygmy hog clusters with ISEA *Sus* and South China pigs, whereas the European pigs and North China wild boars appear to be basal to this cluster. (Fig. 3, Supplementary data 3, Supplementary Figs. 9–15, Supplementary Note). In addition, an ambiguous pattern was also observed in northern Chinese domestic pigs, where the region from 46.2 to 57.1 Mbp support a clustering of northern Chinese domestic pigs with European pigs/ North Chinese wild boars, whereas from 57.1 to 86.8 Mbp support northern Chinese domestic pigs clustering with southern Chinese pigs (Fig. 3, Supplementary Fig. 13). The signature of the introgression regions was also supported by ML trees (Supplementary Fig. 16). Taken together, our results show that within this genomic region, sequences of European/North Chinese pigs have an ancient origin. With the inclusion of pygmy hog genome, we could locate this ghost lineage to be older than the split of pygmy hogs but younger than the split to the Sub-Saharan suids. So far, there is no molecular or fossil evidence for this ancient linage, which was probably extinct long time ago. We further

looked for evidence of this introgression in the autosomes. Comparison between different wild boar populations further identified autosomal regions supporting the X-chromosome introgression signal (Supplementary Figs. 10–15). The amount, length and magnitude of ghost introgression in autosomes are similar among the wild boar populations (Supplementary Figs. 17 & 18), which suggests that this hybridization likely occurred early within the evolutionary history of wild boar.

It has been reported that the region around the centromere on the X chromosome in pigs has an extremely low recombination rate[43,44]. Also, as a consequence of the global reduction in effective population size of wild boar in the past ~1 My to the end of the Last Glacial Maximum[9], wild boar went through processes like incomplete lineage sorting (ILS) and positive natural selection. The joint effect likely resulted in distinct distribution of ancient introgressed haplotype between European/northern Chinese and southern Chinese wild boar populations[40]. Finally, the genealogical discordance on the X chromosome became fixed in pigs from different regions. Recombination rates are highly variable on the autosomes and this hybridization probably happened at least 1 Mya. The long period of recombination, in addition with post-divergence gene-flow between wild boar population[9], highly truncated autosomal haplotype blocks. This would lead to many very short introgression segments scattered over the autosomes, which is what we observed in our analyses (Supplementary Fig. 18).

For the male individuals, we also reconstructed the phylogeny based on the non-recombining part of the Y-chromosome (4.8~43.5 Mb), which resulted in a topology consistent with our main phylogenetic tree (Supplementary Fig. 19).

**Demographical modeling of the Suidae evolutionary history**. We then used an automated qpGraph approach[45,46] to evaluate the fit of various admixture graphs to our data (see Material and methods). We then estimated the marginal likelihood of 37 models that left no f4 outliers using a MCMC approach[47] (Supplementary Fig. 21). According to these models, both pygmy hog and ISEA *Sus* contributed to wild boar. Interestingly, these models suggest that the ISEA *Sus* only contributed to south Chinese wild boar. This discrepancy is likely the result of the near simultaneous divergence of all three wild boar lineages, as well as the fact that a population closer to the South Chinese wild boar potentially also contributed to ISEA *Sus*[21]. The best admixture graph, however, is slightly different from those obtained from *TreeMix* and phylogenetic analyses, as it finds a signal of ISEA *Sus* admixture into pygmy hog population. This suggests that pygmy hog actually interbred with an unsampled admixing/ separating population, which was intermediate between ISEA *Sus* and *Sus scrofa*. However, our parsimonious model may be too simple to reflect the complexity of the reticulate admixture in these populations and to disentangle the ancestral variants between ISEA *Sus* and *Sus scrofa*. Altogether, this analysis validates the existence of gene-flow between wild boar and ISEA *Sus* as well as between pygmy hog and wild boar. Yet, it also suggests that this admixture could have been bidirectional in which case the south Chinese wild boar population forms the best proxy for these events.

To coalesce the known demographic implications by far, we further fitted various fitted various gene-flow models, which are based on our priori assumption of *Suidae* systematic, to a phylogenetic scenario using G-phocs. We separated admixture branches into each of the *Sus scrofa* populations, to better account for their variable levels of basal admixture. (Supplementary Fig. 22, Supplementary Note). In support of our *D*-statistic findings, high probability of gene-flow between the common

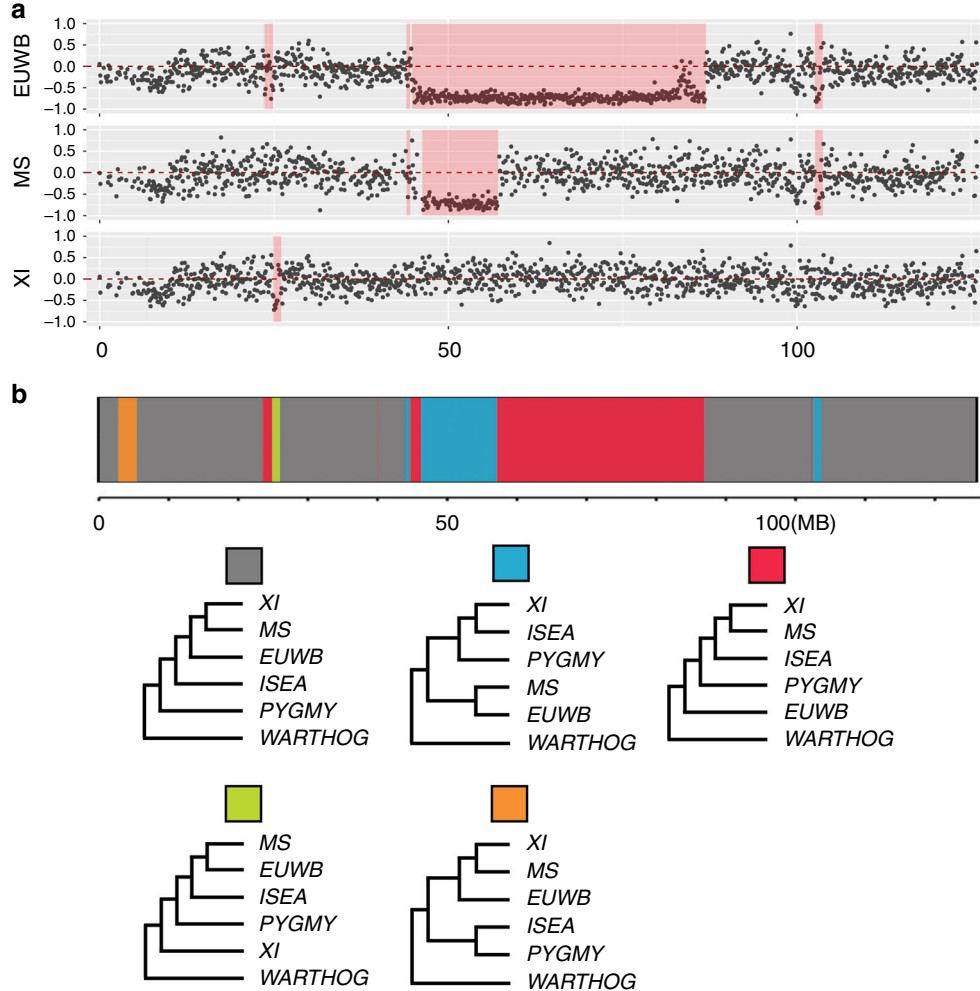

**Fig. 3** Genealogical discordance in Chromosome X. **a** The D-statistic for testing introgression for 100-kb windows on chromosome X for tree topology (((ISEA, X), pygmy hog), warthog) (X = EUWB, MS, and XI). Excesses of BABA in 46–86 Mb indicated higher genetic similarity between ISEA and pygmy hog. See supplementary Figs. 9–14 for D-statistic testing introgression for 100-kb windows on autosome chromosomes. **b** SAGUARO plot illustrating the distribution of the five frequent rooted local topologies over the X chromosome. The red unrooted topologies support introgression from a missing linage into European/North China wild boar. Note that the blue unrooted topology includes the discrepant haplotype within Chinese population (see text for further explanation). ISEA = Islands of Southeast Asia Sus; MS = Meishan; XI = Xiang; EUWB = European wild boar; See Supplementary data 3 for further details and supplementary Figure 8 for results for autosomal chromosomes

ancestor of wild boar and pygmy hog was inferred. In our full model, signatures of admixture in the ISEA population were also examined and significant gene-flow between the ISEA *Sus* and Asian wild boar was found in agreement with previous analyses[21]. Furthermore, a model assuming a basal ghost population was applied and confirmed a post-speciation gene-flow between the common ancestor of wild boar and the ghost population (Supplementary data 4, Supplementary Fig. 23, Supplementary Note).

**Testing for ILS**. It is well known that incomplete ancestral linage sorting can inflate admixture signals. Therefore, we used two methods to test for ILS in our data. First, we calculated calculate the maximum length of a shared haplotype by pygmy hog and wild boar owing to ILS (probability < 0.005). With the deep divergence split between pygmy hog and wild boar (at least 4.2 My), the estimated length of a shared haplotype owing to ILS is extremely small (< 688 bp) and significantly different from the window size used in our sliding-window D-statistic analysis (100 kb). Furthermore, in our Saguaro analysis, we also

filter out all the segment having alternative topology with a length ≤ 688 bp.

Second, we also follow the approach described in Huerta-Sánchez's[27] to assess the probability of ILS. We simulated 10,000 loci with length of 100 kb under the model described in Supplementary Note, and calculated D-statistics with the same quadruplets we used in the sliding-window analysis. All results from the simulations resulted in P < 0.001 against ILS for all comparisons (Supplementary Fig. 25). Thus, we conclude that it is unlikely that ILS have contributed significantly to our observed introgression signal.

**Identification and functional analysis of introgressed genes**. The different introgression signals that we observe in *Sus scrofa* could have played an important role in its successful expansion. We therefore accessed the functional annotation of the genes overlapping introgressed regions. However, given our low-coverage unphased genomic data sets and limited sample size, our ability to reveal ultra-short introgression segments broken down by long-term recombination is limited. With this in mind,

we undertook a functional annotation analysis for candidate introgressed genes. For the introgressed pygmy hog genes in wild boars (384 genes), enrichment for GO terms related to the sensory perception of taste, olfactory pathways and participating in glycolysis and fatty-acid metabolism was observed (Supplementary Fig. 26, Supplementary data 5). This finding is in agreement with the knowledge that smelling, taste, and energy metabolism pathway do have specific roles in adaptive capacity to environment[48–50]. However, it should be noted that especially olfactory genes are prone to copy number variation making them consistent enriched in such analysis. The Ghost introgression genes (104 genes) are related to more broad GO terms including neurogenesis, immune response and TCA cycle (Supplementary data 5). Of the Ghost introgressed genes, the POR gene is of most interest as it is involved in Vitamin D metabolism[51], which could potentially boost the fitness during the expansion of wild boar from sun-belt region (Southern Asia) to short day-length region (Northern Asia and European).

## Discussion

Our analyses reveal the phylogeny and diversification times of the *Suidae* family (Fig. 2). The demographic analysis suggests at least three independent events of inter-species gene-flow during *Suidae* evolution—the most notable from an ancient and now-extinct linage (Fig. 3 and Supplementary Fig. 23). Combined, these results allow us to dramatically refine the evolutionary history of the *Suidae* family. After the Sub-Saharan suids evolved during the late Miocene (~ 10.2 Mya, Fig. 1d), the divergence between pygmy hog and *Sus* took place around the Miocene/Pliocene boundary (~ 6.1 Mya, Fig. 1e), followed by the emergence of ISEA *Sus* and wild boar (Fig. 1f). At ~ 1 Mya, populations of wild boar from Asia started to spread and reach Europe ~ 0.8 Mya (Fig. 1g, h). During this migration, wild boar colonized Eurasia and efficiently replaced all but one of the local species along the way[23,24], with pygmy hog as the only survivor. Moreover, during this the expansion, despite long divergences (~ 2 My between wild boar and pygmy hog, even longer for the ghost linage), wild boar hybridized with both pygmy hogs and an extinct, more divergent, *Suinae* species (Fig. 1h). The frequent climatic fluctuations during the Pleistocene led to alternating warm and cold periods (ices ages)[52,53], which likely resulted in multiple rounds of north-south directed migration[54]. Although expanding instantly, wild boar had greater chances of encountering and temporal co-existing with local species, enabling possible inter-species hybridization. Although our knowledge of the impact of admixture on the fitness of expanding populations is still limited, it is likely that changes in the genetic architecture that arise from admixture will generate heterosis that could boost adaptation to local niches (i.e., high grassland for pygmy hog, high altitude for ghost lineage).

Here, we have shown that an effective replacement of species is accompanied by consistently absorption of part of gene pool of the local related species. This suggests that admixture may play a role as an evolutionary biological driving force in successful range expansion and provides pertinent evolutionary hypothesis on the model of massive species replacement. With the booming development of paleogenomics technology, several case studies have directly verified ancient gene-flow between genomes of extinct species and extent recipient species[27,55]. Future studies, where the genome of fauna from early/middle Pleistocene remain is retrieved, will probably further refine the *Sus scrofa* expansion from Asia to Europe. Overall, the demographic history of pig species not only demonstrates how explosive and invasive range expansion can be, but also reminds us of the ubiquity of inter-species hybridization during speciation.

## Methods

**Sampling, genome sequencing, alignment, and SNP calling.** The pygmy hog used for this research consists of three individuals sampled from the wild and three individuals from captivity. Whole-genome Illumina PE 100 bp re-sequencing was performed at SciGenom Laboratories in Chennai, India on these six pygmy hog samples. The *Babyrousa babyrussa* was sampled from Copenhagen Zoo. Libraries of ~ 300 bp fragments were prepared using Illumina paired-end kits (Illumina, San Diego, CA) and 100 bp paired-end sequenced with Illumina HiSeq. A selection of published genome from other *Suinae* species was included (Supplementary data 1). All these samples were also sequenced with the Illumina sequence technology. The whole-genome sequencing data were trimmed using sickle (https://github.com/najoshi/sickle) with default parameters. The trimmed reads were aligned to the Sscrofa 11.1 reference genome using the Burrows-Wheeler Aligner (BWA 0.7.5a)[56]. Local re-alignment was performed using GATK v3.6.0 RealignmentTargetCreator and IndelRealigner and variants were called using GATK UnifiedGenotyper[57], with the –stand_call_conf option set to 50, the –stand_emit_conf option set to 20, and the –dcov option set to 200. Variants with a read-depth between 0.5 and 2.0 times of the average sample genome coverage were selected and stored in variant calling format. We identified the sex of all individuals by calculating the ratio of read-depth on X chromosome and the autosomes. For the individuals whose molecular sex are male, we filtered out variants in the non-PAR regions, which are heterozygous and with a coverage larger than the average read-depth in autosomes. We do not have any explicit pedigree for the *Babyrousa Babyrussa* sample. To avoid potential biases caused by recent interbreeding, we did decide to used warthog as the outgroup in all the analyses related to introgression. Pygmy hog samples were collected within the Pygmy Hog Conservation Programme in Assam, India in accordance with ethical and legal regulations in India. The *Babyrousa babyrussa* was sampled from a dead individual in Copenhagen Zoo in accordance with ethical and legal regulations in Denmark (The Animal Experimentation Act LBK no. 253, March 8th 2013). This study was ethically approved by the European Research Council under the European Community's 256 Seventh Framework Programme (FP7/2007–2013) / ERC Grant agreement no. 249894".

**Phylogenetic analysis.** Phylogenetic trees on autosomes were construct based on the maximum-likelihood (ML) method as implemented in RAxML 8.2.3[58] using the best-fitting model of substitutions, identified by jModelTest2[59] on 100 random subsets of 1 Mbp. In order to eliminate possible bias stemming from alignment and genotyping errors, we only used autosome one-to-one orthologous gene coding sequences (CDS)[60] between pig and cow for this analysis. A list of one-to-one orthologous genes (between cow and pig) and coordinates of corresponding one-to-one CDS region were extracted from ENSEMBL with biomart[61]. Finally, we got 486,203 CDS regions from 18,313 genes. The total number of SNPs in the one-to-one gene regions was 2571419. We used both supermatrix and supertree techniques[62], using *Babyrousa babyrussa* as an outgroup. In the supermatrix approach, the concatenated CDS alignment was analyzed under best fitting substitution model (GTR + Γ + I) with 100 bootstrap replications. In the supertree approach, pig-to-cow orthologous genes with CDS alignments longer than 1000 bp were used. Individual gene trees were inferred separately under GTR + Γ + I substitution model with 100 rapid bootstraps. All gene trees with an average bootstrap value above 40 were combined into a consensus tree using the software ASTRAL-II[63]. To assess support for particular clades in the supertree analysis, we calculated concordance factors in DensiTree[64].

RAxML 8.2.3 was used to reconstructed ML phylogenetic trees on the whole X chromosome and on the two regions, which have anomalous phylogenetic relationship in the SAGUARO analysis. For mitochondria DNA analysis, we used a Bayesian Markov Chain Monte Carlo simulation (MCMC) to estimate the most likely phylogenetic trees with MrBayes 3.2.3[65], using the best fitting model of substitutions, identified by JModelTest2. The length of the MCMC was set to 10,000,000. The parameter estimates and consensus trees resulting from 10 MrBayes runs were recorded and compared. The best-supported phylogenetic consensus tree was summarized with TRACER v1.6 (http://tree.bio.ed.ac.uk/software/tracer/) discarding the first 10% as burn-in. All trees were depicted using the software FigTree v1.4.2 (http://tree.bio.ed.ac.uk/software/figtree/).

**Molecular clock analyses.** We estimated divergence times using an approximate likelihood method as implemented in MCMCtree[66], with an independent relaxed-clock and birth-death sampling. To overcome difficulties arising from computational efficiency and admixture, we only used CDS with pig-to-cow ortholog filter. We fitted an GTR + Γ4 model to each genomic bin and estimated a mean mutation rate by fitting a strict clock to each fragment setting a root age at 20 Mya, which represent the earliest *Tayasuidae* fossil. This mean rate was used to adjust the prior on the mutation rate (rgene) modeled by a gamma distribution as G (1,241). The BDS and sigma2 values were set at 7 5 1 and G (1,10) respectively. We ran 100,000 (25% burn in) MCMC samples for fossil calibration reported previously. We used a float prior and a maximum bound age, with a scale parameter of $c = 2$. Allowing the MCMC to explore a wide range of time for the divergence between *Babyrussa* and *Suinae* and calibrate the time later than the split of "new world" peccaries (tU = 2 [20 Mya], $p = 0.1$, $c = 2$). For MRCA of sub-Saharan African suid and *Sus*, we used the same fossil calibration as in Frantz et al. 2013 (tL = 0.55 [5.5 Mya], $p = 0.9$, $c = 0.5$). For MRCA of *Sus*, we used a minimum bound at 2 Ma [tL = 0.2 [2

Mya], $p = 0.1$, $c = 0.5$] to represents the earliest appearance of *Sus* in the fossil record of Island Southeast Asia.

**Detecting gene-flow among *Suidae*.** We integrated Patterson's *D*-statistic to examine the phylogenetic distribution of derived alleles at loci that display either an ABBA or BABA allelic configuration across the chromosomes among *Suidae* using warthog as an outgroup. For admixture estimation, we assigned 18 autosomes and selected a block size of 5 Mb to calculate the standard errors on *D*-statistics using Admixtools[45]. We also identified candidate introgression loci using *D*-statistic and fd in slide window[67]. To avoid D returning inflated values in small genomic regions[68], we set the window size as 100-Kb and summarized the results in Venn diagrams.

**SAGUARO.** Phylogenetic relationships of genomic regions may differ from the species tree due to incomplete linage sorting and introgression. To test whether this is the case in our analysis, and to detect breakpoints between genomic segments supporting different local topologies, we used the machine-learning approach implemented in SAGUARO[42]. We first ran SAGUARO with six representative individuals for the overview of whole genome (See Supplementary data 3). Then, to better estimate length of phylogenetic incongruent regions, we performed the same approach but using the quadruplets in sliding-window *D*-statistic analysis. We constrained SAGUARO to use only nucleotide positions with no missing data and ran with 20 iterations and 500 neurons.

**Fitting models of population history.** We used qpGraph[45] to fit admixture graphs to six populations representing European wild boar, North Chinese wild boar, South Chinese wild boar, ISEA, pygmy hog, and warthog as the outgroup. We filter the data set using following criteria: SNPs with at least 10 Kb distance from one another, no more than 10% missing data. This resulted in 361,837 SNPs. To explore the space of all possible admixture graphs, we used a heuristic search algorithm first described in Leathlobhair et al.[46] (code available at https://github.com/ekirving/qpbrute). Given an outgroup with which to root the graph, a stepwise addition order algorithm was used for adding leaf nodes to the graph. At each step, insertion of a new node was tested at all branches of the graph, except the outgroup branch. Where a node could not be inserted without producing f4 outliers (i.e., | Z| ≥ 3) then all possible admixture combinations were also attempted. If a node could not be inserted via either approach, that sub-graph was discarded. If the node was successfully inserted, the remaining nodes were recursively inserted into that graph. All possible starting node orders were attempted to ensure full coverage of the graph space. We fitted 2444 unique admixture graphs for these six populations and recorded the 37 graphs that left no f4 outliers (i.e., |Z| < 3). We then used the MCMC algorithm implemented in the ADMIXTUREGRAPH R package[47] to compute the marginal likelihood of these 37 models and their Bayes Factors (BF). We ran two independent replications, each with two million iterations, five heated chains, a burn in of 50%, and no thinning. Convergence was assessed by calculating the potential scale reduction factor for the model likelihoods using the CODA R package[69]. We found one particularly well supported model (1d9676e) which, when compared with all others, had K < 119 (Supplementary Figs. 20–21). We also note that among the remaining models there are small differences in the admixture topologies, however, most models support gene-flow between ISEA *Sus* and Chinese pigs, as well as between the pygmy hog and basal wild boar (Supplementary Fig. 21).

We further carried out demographic analysis based on the G-PhoCS, applied to 10,000 loci of 1 kb of length chosen via a series of filter to obtain putatively neutral loci. We filtered out exons of protein coding genes and 10 kilobases (kb) flanking them on each side, as well as conserved noncoding elements and 100 bases on each side of these elements. We selected 1 kb loci located at least 30 kb apart. We identified a collection of 11,274 loci that followed these criteria, and random selected 10,000 loci for the G-PhoCS. Multiple sequence alignments for these loci were extracted using sequence data from the all individual genomes. We conditioned inference on the population phylogeny based upon the neighbor-joining tree constructed with MEGA[70] on the basis of the IBS distance matrix data of neutral loci used in G-PhoCS analysis calculated by PLINK 1.9[71] (Supplementary Fig. 24). The prior distributions over model parameters was defined by a product of Gamma distributions with $\alpha = 1$ and $\beta = 10,000$ for population size and divergence time scaled by mutation rate, and $\alpha = 0.002$ and $\beta = 0.00001$ for the migration rates. Markov Chain was run for 100,000 burn-in iterations, after which parameter values were sampled for 200,000 iterations every 10 iterations, resulting in a total of 20,001 samples from the approximate posterior. Convergence was inspected manually for each run (effective sample size for all parameters > 200). We converted probabilities into rates using the formula $p = 1-e^{-m}$ (where *p* is the probability of gene-flow and *m* is the total migration rate)[72]. We checked for convergence between runs using Tracer v1.7[73].

Finally, we also used Treemix v1.12[74] to test models of possible admixture for Babyrussa, warthog, pygmy hog, ISEA, European pigs, Northern China pigs, and Southern China pigs. Windows with 500 consecutive SNPs were used to account for the non-independence of SNPs located in close vicinity. Migrations from m0 to m10 were tested, with five replicates per m to assess consistency.

**Probability of introgression fragment from shared ancestral lineage.** Based on the equation described in Huerta-Sánchez et al.[27], we calculate the probability of a haplotype shared by pygmy hog and wild boar as a result of ancestral ILS. In brief, let *k* be the introgressed haplotype length of the two species' branches since divergence. The expected length of a shared ancestral sequence is $L = 1/(r \times t)$, where *r* is the recombination rate and *t* is branch lengths of pygmy hog and wild boar since divergence. The probability of a length of at least *k* is 1-GammaCDF(*k*, shape = 2, *L* = 1/*L*), where GammaCDF is the Gamma distribution function. The lower estimate of 4.2 My of the Sus–pygmy hog was used as branch length and an assumed generation time of 5 years. The recombination rate was set to 0.8 cM/Mb[75].

Another approach to assess probability of ILS is comparing *D*-statistics between populations under simulations of demographic model. However, it requires a very detailed and precise demographic model to obtain a better assessment. The historical demographic information of pygmy hog and the ancestral population of Suinae species are still deficient. Here, we can only fit in a simplified model. Inaccuracy of the effective ancestral population size and bottleneck event may lead to over-/underestimation of ILS. With this in mind, we compared *D*-statistics with the same quadruplet as we used in the sliding-window steps under simulations of a simple demographic model with no gene-flow. (see Supplementary Note for details). All simulations resulted in *P* < 0.001 against ILS for all comparisons (see Supplementary Fig 25 and Supplementary Note).

**Functional annotation of genes in introgressed regions.** We applied a functional annotation analysis using PANTHER v.11[76] on the candidate introgressed genes. Genes from the pygmy hog/Sus scrofa introgression and the Sus scrofa/ghost lineage introgression were analyzed separately. Gene-enrichment analyses were performed using clusterProfiler[77]. False discovery rate was performed to adjust *P* values using the Benjamini and Hochberg method. A *P* value of < 0.05 was used as the cutoff criterion.

## Data availability

The authors declare that all data supporting the findings of this study are available within the article and its Supplementary Information files, or from the corresponding author upon request. Raw reads of pygmy hog and *Babyrousa babyrussa* have been deposited in the European Nucleotide Archive (ENA) under accession PRJEB30129. Sequences for the sequenced *Sus scrofa* and other species have been deposited on the EBI Sequence Read Archive under accession number ERP001813.

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

## Acknowledgements

L.L. has received financial support from the China Scholarship Council (Grant No. 201707720055). This project was financially supported by a European Research Council grant (ERC-2009-AdG: 249894). Sample of *Babyrousa babyrussa* was kindly provided by professor Dr. M. Fredholm, University of Copenhagen.

## Author contributions

M.A.M.G., O.M., and L.L. designed the study. M.A.M.G., O.M., and H.-J.M. initially conceived the project; G.N. provided the pygmy hog samples; L.L. analyzed the data; Y.-L.L. and L.L. performed the phylogenetic analyses; E.K.I.-P. preformed the qpgraph analyses; L.L., M.B., H.-J.M., and O.M. discussed the results; L.L. wrote the manuscript; M.B., H.-J.M., L.A.F.F., M.A.M.G., and O.M provided valuable suggestion and comments to improve the manuscript.

## Additional information

**Competing interests:** The authors declare no competing interests.

