## [Peer Review File · Nature Communications]

Reviewers' comments:

Reviewer #1 (Remarks to the Author):

Elucidating evolutionary history of pig (*Sus scrofa*) as well as their related species increase our cognition to pigs, provide insights to the development of two foundational functions of pigs (i.e., a main source of animal protein and serves as a biomedical model for human disease). The MS entitled "The little Pygmy hog (*Porcula salvania*), a big piece in resolving the complex Suidae speciation puzzle" reported the whole-genome sequences of six pygmy hog individuals (the only suid species) and one *Babyrusa babyrussa* individual. Combined with publicly available 31 suid genomes, Liu et al. performed a phylogenetical analysis of 10 of the extant Suidae species and demonstrated the rapid spread of wild boar across Eurasia and North Africa was likely facilitated by inter-specific/inter-generic admixtures, instead of the replacement of the species it encountered. This is a significantly advanced in rounding out the complex demography and evolutionary history of Suidae species.

I appreciate authors' efforts for initiating this effort in pygmy hog, which have the potential to produce significant insights into the evolution and biology of Suidae species. The quality of the presentation is excellent, the structure of the presentation is clear and there are a very small number of typographical errors.

Overall the conclusions appear sound and reliable, nonetheless, the relatively lower coverage depth for six pygmy hogs (five with ~ 11 x coverage and one with ~ 31 x coverage) and one *Babyrusa babyrussa* individual (~ 11 x coverage) is clearly the Achilles heel of this study. Logically, the accuracy and number of identified SNPs were highly correlated with coverage depth.

Equally important, assembly of the reference pig genome (Sscrofa10.2) used in this study is largely incomplete (at least 8% of the sequence is estimated to be missing from the assembly), I strongly suggest the author should use the updated high-quality assembly of the reference pig genome (Sscrofa11.1), which released at Dec 2016 (Contig N50 = 48.23 Mb, scaffold N50 = 88.23 Mb, 22,452 annotated protein coding genes). The Sscrofa11.1 is substantially superior than the former version of Sscrofa10.2 (Contig N50 = 69.67 Kb, scaffold N50 = 576.01 Kb, and 21,630 annotated protein coding genes).

Especially for the X chromosome, which been separately analyzed with the autosomes in this study, compared with the Sscrofa10.2, many gaps have been filled and the order of sequences on the chromosome in the updated assembly (Genome Res. 2016. doi: 10.1101/gr.188839.114). Many genes from the Sscrofa10.2 (422 protein coding genes) that were disrupted by gaps have now been completed in Sscrofa11.1 (690 protein coding genes). I think these improvements should be substantively done to strength the robustness of the conclusions drawn.

Specific comments:

1) I think the title should accurately highlight the core scientific discovery of the study, rather than the non-technical description as "a big piece in resolving the complex puzzle".

2). Genome sequencing, alignment and SNP calling

Given a deep phylogenetically split between pygmy hog (*Porcula salvania*) and pig (*Sus scrofa*), it is necessary and meaningful to provide the mapping statistics of the whole genome re-sequencing data for each individual. Typically, this summary should include the high-quality ratio (%), mapping ratio (%), coverage of genome at 4x depth (%) and so on.

The number of SNPs is highly correlated with sequencing depth. As shown in Table S1, although six pygmy hog individuals had been shotgun sequenced with Illumina's technology but at highly unbalanced depths, ranging from ~ 9.26 x for individual "LIB5206_59F4201" to 31.2x for individual "LIB5207_60F4192". The author did not provide any explanation about the correction of the bias resulted from the different depth and make them are comparable. The author also should provide the statistics and the annotations of the identified SNPs.

3) Lines 87-88: "We sequenced the genomes of six pygmy hogs and one *Babyrusa babyrussa* and analyzed these with the genome sequences of 31 individuals". I have not find the corresponding

description of the *Babryrousa babyrussa* sample in section of "Materials and methods", which is sampled and sequenced in this paper or not?

4) Fig. 3: The author reconstructed a maximum-likelihood tree for the X chromosome, nonetheless, the sex information of each individual were not provided. The author should pay attentions and further confirm the accuracy of the SNPs in X chromosome for the males, especially given that the relatively lower coverage depth in this paper.

If the male individuals are available, I also believe there is an opportunity to explore the Y-chromosome variations. More recently, Guirao-Rico et al. reported the SNP variations in porcine Y-chromosome is consistent with the occurrence of paternal gene flow from non-Asian to Asian populations (Heredity, 2018, doi.org/10.1038/s41437-017-0002-9).

5) Although the author found the rapid spread of wild boar across Eurasia and North Africa was likely facilitated by inter-specific/inter-generic admixtures, I also believe there is an opportunity lost here. The explore of the role of introgression in adaptation will makes this study is more biologically meaningful and greatly enhance the scientific priority of this paper. More recently, Ai et al found the adaptive sweep in the high-latitude regions has acted on DNA that might have been introgressed from an extinct *Sus* species. I think this previous report provides a good paradigm to refer.

6) Lines 248-251: 'In order to eliminate possible bias stemming from alignment and genotyping errors, we only used autosome one-to-one orthologous gene coding sequences (CDS) between pig and cow for this analysis.' I have not found the details about the identification of single copy orthologous between pig and cow. Given the volume of information available, it is difficult to assess the methodology.

7) Lines 74-75: "Under this scenario, wild boar absorbed rather than replaced species, a process paralleled to some extent in humans^{27,28}." I suggested the more recently paper 'The genome of the offspring of a Neanderthal mother and a Denisovan father, Nature 561, 113–116 (2018)' should be cited.

8) The references 40 and 42 are repetitive, as well as 9 and 46.

9) Accession codes: Raw reads of pygmy hog and *Babryrousa babyrussa* have been deposited in the European Nucleotide Archive (ENA) under accession *** (accession number will be available when paper is accepted). In my opinion, the sequencing data should be deposited in database before a manuscript describing the data is sent to a journal for review, and the temporary "reviewer's link" should provided for the peer review purpose only.

Reviewer #2 (Remarks to the Author):

This is a very interesting paper that should reach a wide audience. The study provides evidence for the role of inter-species hybridization in the evolutionary success of a species; the wild boar is a striking example of such phenomenon. The paper is clear and well written and the conclusions rely on a large amount of appropriate analyses.

One point which would make the study even better and increase its significance is the assessment of the adaptive value of the introgressive hybridization. The authors state that (lines 175-77) this is limited by the inability to identify the short introgression segments. However, it should be possible to identify, at least for a part of the regions, genes likely to be impacted. Even if advantageous alleles may not be identified precisely, this would bring insight in the functions affected by these introgressions. Otherwise, the authors should provide more precise arguments to demonstrate that this is not technically possible.

Besides that, I have a few small concerns:

- several samples were collected in zoos. Is there any information on their genealogy/origin with

regard to the prevention of inter-specific crosses that would affect the results?

- at first reading, it is not easy to find the way among all the different pig groups in both the main text and figures. To make this more straightforward, I would create in the Material and Methods a "Sampling" section different from the "Sequencing, alignment and calling" section, and complete Suppl. Tab 1 with a column "Species name" (distinct from the breed name) and a column with the category code used (ISEA, EUD, EUWB, etc...). I would also give the meaning of these codes in the caption of Fig 2 so that it would be self-sufficient.

- The authors should be more explicit (lines 152-155) in detailing the arguments supporting a 'ghost' origin of the introgression.

- It would be relevant to have an idea of the amount of data used for the study (line 250): how many CDS does this represent? And how many related SNPs?

Minor points / typos:

- line 248: how many random subsets used ?
- line 262 "mitochondria" → mitochondrial DNA
- line 284 Babyrusa → italic
- line 298 "summarize" → summarized
- line 303 "linage" → lineage
- Refs nr 40 and 42 are identical.
- Supplementary material: in many places "out group" → outgroup
- Suppl Fig 3. "Mitochondrial phylogeny" → Phylogeny based on mitochondrial DNA
- Suppl. Fig 18. (i) title is lacking. (ii) " This figure only show" → "This
- Suppl. Note: "From this neutral dataset, we first constructed a NJ tree which is concordance with the main topology of the used population (Supplementary Fig. 19)." "concordance" → concordant

Reviewer #3 (Remarks to the Author):

I appreciate that the authors have sequenced the genomes of pygmy hog, an important species for studying evolution of Suidae. The authors have done some analyses to understand the phylogeny, admixture of pygmy hog and wild boar. I think they have made some discoveries, but it is not novel, and the analyses are simple. In addition, it is way too technical for anything but a specialist in the field to read. I think this paper is better suited to Molecular Biology and Evolution or Molecular Ecology, rather than Nature communications.

My major comments:

1. Section of "Phylogenetic relationships and divergence of the Suidae species"

As X chromosome evolves with introgression, I suggest the authors to present the results of phylogenetic tree by autosome, X and independently, and discuss potential difference and underlying confounding factors.

2. I strongly suggest the authors to use more software to infer divergence time. As i know, some discrepancies exist by different methods. The authors should keep in mind that the divergence time might be short using strategy of variant calling by re-sequenced genomes

3. One of my major concerns is the shortcoming of D-statistics, which the author used to infer the

signal of genetic introgression. As we know, mapping re-sequenced genomes, particularly from another divergent species (pygmy hog), to the reference genome of another species (pig), will generally miss many mutations.

I strongly suggest three analyses: 1) sequence simulation with the information of mutation rate and divergence time. You likely find the false positive signal of introgression using short read mapping strategy for variants calling, even without genetic introgression. 2) trying to extract sequence directly from BAM alignment file for pygmy hog, or assemble sequence and align assembly to reference to call variants. 3) including much more different methods to validate admixture, such as Treemix, RFMIX, fd.....

Again, using D-statistics to infer regions of introgression will likely generate high false positive. At least, phylogenetic trees constructed using the candidate introgressed regions are necessary to validate the signal of introgression. I suggest the authors to use S^* used in human, and rIBD described in Bosse et al 2014.

Many homologous regions that are challenge for variant calling will be commonly presented, to actually false positive.

4. Is there any reference reporting that pygmy hog can hybrid with pig/wild boar?

5. excluding the possibility of incomplete lineage sorting as described in Huerta-Sánchez et al (2014, nature) is necessary to infer admixture.

6. The authors sequenced the genomes of Duroc, Large white, Meishan, Xiang, but it is unclear why the authors chose them for sequencing. Using one individual of Duroc, Large white, Meishan, Xiang, to represent European, northern Chinese and southern Chinese pigs is dangerous.

There are many released high coverage genomes of domestic pig and wild boar across Europe and China, e.g. Ai et al 2014 (Nat genet), Leno-Colorado et al (G3 2017 7(7):2171-2184). Why not use these for analysis? In addition, I am afraid that Meishan is not a Northern Chinese breed. The most famous Northern Chinese breed is Min.

7. line 13-140. Description in ref 42: " we found an exceptionally large (14-Mb) region with a low recombination rate on the X chromosome that appears to have two distinct haplotypes in the high- and low-latitude populations, possibly underlying their adaptation to cold and hot environments, respectively." No South Asia pig was studied in this paper.

8. Section "Wild boar harbors genetic introgression from a ghost lineage." The finding of X chr introgression has been described in Ai et al (2014). I don't think there is any important discovery here.

9. Figure 3b, why using unrooted tree? It is difficult to understand the relationship using unrooted tree.

Dear Editor and Reviewers:

We would like to thank the three reviewers for their positive and constructive comments and suggestions which have been very helpful in revising and improving our manuscript. We have carefully revised our manuscript after reading the comments and below we provide point to point responses (in italic) to reviewers' comments (in bold). The major changes of the manuscript are that we repeated all analyses based on the latest reference pig genome (Sscrofa11.1) and we also conducted additional analyses to specifically assess the authenticity of the inter-species gene flow signal (e.g. ILS versus admixture). Revised portions are marked in red in the manuscript.

Reviewer #1:

- 1) Overall the conclusions appear sound and reliable, nonetheless, the relatively lower coverage depth for six pygmy hogs (five with ~11 x coverage and one with ~31 x coverage) and one *Babirusa babirusa* individual (~11 x coverage) is clearly the Achilles heel of this study. Logically, the accuracy and number of identified SNPs were highly correlated with coverage depth.**

We agree that the number of SNPs to some extent is correlated with sequencing depth. In the manuscript, we have now included statistics for mapping and identifying SNPs (Sup table 1). From the statistics we can see that difference in depth does not lead to a significant difference in the number of identified SNPs between the six pygmy hogs. Furthermore, for the analysis where we only included one pygmy hog sample, we always used the sample with the highest coverage in order to use the most reliably called SNPs. Thus, although we do acknowledge that some bias will result from the differences in sequence coverage of our dataset, we are confident that the criteria used to call SNPs have minimized this bias and do not significantly influence the main finding of the paper.

- 2) Equally important, assembly of the reference pig genome (Sscrofa10.2) used in this study is largely incomplete (at least 8% of the sequence is estimated to be missing from the assembly), I strongly suggest the author should use the updated high-quality assembly of the reference pig genome (Sscrofa11.1), which was released in Dec 2016 (Contig N50 = 48.23 Mb, scaffold N50 = 88.23 Mb, 22,452 annotated protein coding genes). The Sscrofa11.1 is substantively superior to the former version of Sscrofa10.2 (Contig N50 = 69.67 Kb, scaffold N50 = 576.01 Kb, and 21,630 annotated protein coding genes). Especially for the X chromosome, which was separately analyzed with the autosomes in this study, compared with the Sscrofa10.2, many gaps have been filled and the order of sequences on the chromosome in the updated assembly (Genome Res. 2016. doi: 10.1101/gr.188839.114). Many genes from the Sscrofa10.2 (422 protein coding genes) that were disrupted by gaps have now been completed in Sscrofa11.1 (690 protein coding genes). I think these improvements should be substantively done to strengthen the robustness of the conclusions drawn.**

We totally agree with reviewer 1 that the new pig reference genome assembly (Sscrofa11.1) is in many aspects substantially superior to the previous pig reference genome (Sscrofa10.2).

Since some of our analyses in our original manuscript submission were done before the release of Sscrofa11.1, we had decided to present all the analysis/results based on Sscrofa10.2 as we assumed that the main conclusions would not change by using the new reference genome. In the revised manuscript, we have now mapped all data against the new reference and all analyses are based on the alignments against Sscrofa11.1. In the revised version of the manuscript, we have adapted the materials/methods and result sections accordingly. The new results are highly consistent with the original results with only minor differences which generally strengthen our conclusions. For example, in the X chromosome introgression analysis, using the new genome build, we have been able to identify the coordinate of the introgressed segment with more accuracy and we also obtain fewer alternative supported topologies.

- 3) I think the title should accurately highlight the core scientific discovery of the study, rather than the non-technical description as “a big piece in resolving the complex puzzle”.**

We see the point of the reviewer but we feel that the title does cover quite well the merit of the paper as pygmy hog, in our opinion, is an important puzzle piece in understanding the complex speciation puzzle of Suidae including the massive range expansion of Sus scrofa which is part of this fascinating speciation puzzle. We have therefore decided to keep the title and hope that the reviewer accepts our argumentation.

- 4) Genome sequencing, alignment and SNP calling**

Given a deep phylogenetically split between pygmy hog (*Porcula salvania*) and pig (*Sus scrofa*), it is necessary and meaningful to provide the mapping statistics of the whole genome re-sequencing data for each individual. Typically, this summary should include the high-quality ratio (%), mapping ratio (%), coverage of genome at 4x depth (%) and so on.

The number of SNPs is highly correlated with sequencing depth. As shown in Table S1, although six pygmy hog individuals had been shotgun sequenced with Illumina’s technology but at highly unbalanced depths, ranging from ~ 9.26x for individual “LIB5206_59F4201” to 31.2x for individual “LIB5207_60F4192”. The author did not provide any explanation about the correction of the bias resulted from the different depth and make them are comparable. The author also should provide the statistics and the annotations of the identified SNPs.

See our reply to point number one.

Lines 87-88: “We sequenced the genomes of six pygmy hogs and one *Babyrousa babyrusa* and analyzed these with the genome sequences of 31 individuals”. I have not find the corresponding description of the *Babyrousa babyrusa* sample in section of “Materials and methods”, which is sampled and sequenced in this paper or not?

**Babyrousa babyrusa* is indeed a new dataset. In the revised manuscript, we added the description of sampling and sequencing for *babyrusa* sample (line 292-294 and Sup table 1).*

- 5) Fig. 3: The author reconstructed a maximum-likelihood tree for the X chromosome,**

nonetheless, the sex information of each individual were not provided. The author should pay attentions and further confirm the accuracy of the SNPs in X chromosome for the males, especially given that the relatively lower coverage depth in this paper. If the male individuals are available, I also believe there is an opportunity to explore the Y-chromosome variations. More recently, Guirao-Rico et al. reported the SNP variations in porcine Y-chromosome is consistent with the occurrence of paternal gene flow from non-Asian to Asian populations (Heredity, 2018, doi.org/10.1038/s41437-017-0002-9).

For some samples the gender record is missing, thus we applied a molecular sex determination by calculating the ratio of read depth on the X chromosome and the autosomes. The result is showed in Sup Figure 7. For the individuals whose molecular sex was assigned as male, we filtered out variants on the X-chromosome which were heterozygous and variants with coverage larger than the average read-depth in autosomes.

For the Y chromosome, we reconstructed the phylogeny for the nonPAR regions in male individuals. The Y chromosome phylogeny is consistent with the species tree topology, thus we did not observe any genealogical discordance. Since we do not have any male wild boar individual from Southern China in our dataset, unfortunately, we cannot validate the finding from Guirao-Rico et al.

- 6) **Although the author found the rapid spread of wild boar across Eurasia and North Africa was likely facilitated by inter-specific/inter-generic admixtures, I also believe there is an opportunity lost here. The explore of the role of introgression in adaptation will makes this study is more biologically meaningful and greatly enhance the scientific priority of this paper. More recently, Ai et al found the adaptive sweep in the high-latitude regions has acted on DNA that might have been introgressed from an extinct Sus species. I think this previous report provides a good paradigm to refer.**

We agree that evaluating the functional role of introgression will enhance the scientific value of the manuscript. During the analysis phase for the paper, we did perform functional enrichment analyses, but finally decided to leave it out of the paper to keep a clearer focus on the role of introgression during speciation and range expansion. We have now included gene enrichment analysis on the genes overlapping introgression regions and the results are included in the results section line 228-247 and a full gene list can be found in Supplementary Table 5. We did not find any enrichment specifically related to adaptation which does not exclude that these genes could be involved in adaptation as adaptive introgression does not necessarily result in functional gene enrichment.

- 7) **Lines 248-251: 'In order to eliminate possible bias stemming from alignment and genotyping errors, we only used autosome one-to-one orthologous gene coding sequences (CDS) between pig and cow for this analysis.' I have not found the details about the identification of single copy orthologous between pig and cow. Given the volume of information available, it is difficult to assess the methodology.**

This was indeed not well described. One-to-one orthologous genes (between cow and pig) and coordinates of corresponding one-to-one CDS region were extracted from ENSEMBL with

biomart. In the revised manuscript we added the methodology in Materials and Methods section line 316-319.

- 8) **Lines 74-75: “Under this scenario, wild boar absorbed rather than replaced species, a process paralleled to some extent in humans^{27,28}.” I suggested the more recently paper ‘The genome of the offspring of a Neanderthal mother and a Denisovan father, Nature 561, 113–116 (2018) ’should be cited.**

We agree that this up to date paper is more appropriate here and we now have included a reference to this paper (ref 29).

- 9) **The references 40 and 42 are repetitive, as well as 9 and 46.**

We are happy that the reviewer pointed out these redundancies in our references. In the revised manuscript we have carefully checked all the citations.

- 10) **Accession codes: Raw reads of pygmy hog and *Babryrousa babyrussa* have been deposited in the European Nucleotide Archive (ENA) under accession *** (accession number will be available when paper is accepted). In my opinion, the sequencing data should be deposited in database before a manuscript describing the data is sent to a journal for review, and the temporary “reviewer’s link” should provide for the peer review purpose only.**

We have deposited the new data to ENA under accession number PRJEB30129. We are following the data availability policy of Nature communication stating that the data must become available upon publication. We have asked ENA for a “reviewer link” as suggested by the reviewer but ENA unfortunately does not provide such a “reviewer link”

Reviewer #2:

- 1) **One point which would make the study even better and increase its significance is the assessment of the adaptive value of the introgressive hybridization. The authors state that (lines 175-77) this is limited by the inability to identify the short introgression segments. However, it should be possible to identify, at least for a part of the regions, genes likely to be impacted. Even if advantageous alleles may not be identified precisely, this would bring insight in the functions affected by these introgressions. Otherwise, the authors should provide more precise arguments to demonstrate that this is not technically possible.**

See answer to reviewer 1 point 6

- 2) **Several samples were collected in zoos. Is there any information on their genealogy/origin with regard to the prevention of inter-specific crosses that would affect the results?**

As far as we are aware there are no records on possible inter-specific crosses in the used zoo samples.

- 3) **At first reading, it is not easy to find the way among all the different pig groups in both the main text and figures. To make this more straightforward, I would create in the Material and Methods a “Sampling” section different from the “Sequencing, alignment and calling” section, and complete Suppl. Tab 1 with a column “Species name” (distinct from the breed name) and a column with the category code used (ISEA, EUD, EUWB, etc...). I would also give the meaning of these codes in the caption of Fig 2 so that it would be self-sufficient.**

This is indeed a good suggestion and we have therefore adapted this following the Reviewer's suggestion.

- 4) **The authors should be more explicit (lines 152-155) in detailing the arguments supporting a ‘ghost’ origin of the introgression.**

In the revised manuscript, we have added a more detailed interpretation of the unknown Suidae species. (line 157-160)

- 5) ***It would be relevant to have an idea of the amount of data used for the study (line 250): how many CDS does this represent? And how many related SNPs?***

In the revised manuscript, we have added statistics of the CDS region. (line 316-319)

- 6) **Minor points / typos**

For the minor points / typos, we have corrected all of these according to the Reviewer's comments. We thank the reviewer for pointing those errors out.

Reviewer #3:

- 1) **I appreciate that the authors have sequenced the genomes of pygmy hog, an important species for studying evolution of Suidae. The authors have done some analyses to understand the phylogeny, admixture of pygmy hog and wild boar. I think they have made some discoveries, but it is not novel, and the analyses are simple. In addition, it is way too technical for anything but a specialist in the field to read. I think this paper is better suited to Molecular Biology and Evolution or Molecular Ecology, rather than Nature communications.**

*We respectfully disagree with this comment. We believe our study of pygmy hog is insightful given the unprecedented accuracy of phylogeny, divergence time and demographic history of the Suidae family. Also, we found that wild boars have had at least three independent post-speciation admixtures. Although the genealogical discordance on the X chromosome has been reported in previous studies (e.g. Tortereau, et al. BMC genomics. 2012 Dec;13(1):586; Ai, et al. Nature genetics. 2015 Mar;47(3):217.), with our newly presented complete genome sequences of pygmy hogs and *Babyrussa babyrussa*, we are able to more precisely locate the phylogenetic placement of the introgression donor. We agree with the two other reviewers that “...This is a significantly advance in rounding out the complex demography and evolutionary history of Suidae species. I appreciate authors' efforts for initiating this effort in pygmy hog, which have the potential to produce significant insights into the evolution and biology of Suidae*

species. The quality of the presentation is excellent, the structure of the presentation is clear and there are a very small number of typographical errors....” (reviewer #1) and “...This is a very interesting paper that should reach a wide audience. The study provides evidence for the role of inter-species hybridization in the evolutionary success of a species; the wild boar is a striking example of such phenomenon. The paper is clear and well written and the conclusions rely on a large amount of appropriate analyses...” (reviewer #2). Our results not only demonstrate how explosive and invasive range expansion can be, but also emphasize the ubiquity of inter-species hybridization during speciation. Our study results provide a panorama of the evolutionary history of Suidae species to other investigators, thereby helping to pave the way for further research into elucidating the evolutionary biological driving force enabling successful range expansion. Thus, we consider the significance and impact of our conclusions to be worthy of publication in a high profile, non-specialist academic journal and thus believe that our manuscript is appropriate for publication by Nature Communications.

Section of "Phylogenetic relationships and divergence of the Suidae species". As X chromosome evolves with introgression, I suggest the authors to present the results of phylogenetic tree by autosome, X and independently, and discuss potential difference and underlying confounding factors.

We now also have reconstructed the phylogenetic tree of the X chromosome (Sup Figure 8). Since we already have prior knowledge that there is a large region on the X chromosome showing strong genealogical discordance, the tree based on the complete X chromosome does not represent the evolutionary history of the X chromosome, but instead represent the strongest phylogenetic signal present in the X chromosome. Thus, to deal with this discordance, we instead applied a sliding-window D-statistic (Saguaro et al) to further investigate the complicated history of the X chromosome.

2) I strongly suggest the authors to use more software to infer divergence time. As i know, some discrepancies exist by different methods. The authors should keep in mind that the divergence time might be short using strategy of variant calling by re-sequenced genomes

To our knowledge, we do not know of any other time of divergence software which can handle whole genome datasets and we also believe that different time of divergence software will result in highly similar divergence times. Based on a comparison study of different molecular clock dating strategies Barba-Montoya et al. (Molecular phylogenetics and evolution. 2017 Sep 30;114:386-400.), showed that truncation and effective priors have considerable impact on calibrations. With a proper prior setting, different methods should have similar results. In our study, MCMCtree settings and calibration settings are adapted from our previous study (Frantz et al. Genome Biology, 2013) where extensive in-depth analysis was applied to find the proper prior settings. Here, we random sample 5 Mb CDS sequence and run a BEAST analysis with the exact same prior. BEAST analysis resulted in similar divergence time with MCMCtree (See figure below). Due to the computational issues, MCMCtree is an approximate likelihood method which can handle large datasets. We therefore think that

our current time of divergence estimates are the best estimates that can be achieved with the current available time of divergence software that can handle whole genome data

Fig. Estimation of divergence times obtained from BEAST v1.8. 95% HPD intervals are shown as blue bars. The 95% HPD is regarded as a Bayesian representation of confidence interval.

3) One of my major concerns is the shortcoming of D-statistics, which the author used to infer the signal of genetic introgression. As we know, mapping re-sequenced genomes, particularly from another divergent species (pygmy hog), to the reference genome of another species (pig), will generally miss many mutations. I strongly suggest three analyses: 1) sequence simulation with the information of mutation rate and divergence time. You likely find the false positive signal of introgression using short read mapping strategy for variants calling, even without genetic introgression. 2) trying to extract sequence directly from BAM alignment file for pygmy hog, or assemble sequence and align assembly to reference to call variants. 3) including much more different methods to validate admixture, such as Treemix, RFMIX, fd..... Again, using D-statistics to infer regions of introgression will likely generate high false positive. At least, phylogenetic trees constructed using the candidate introgressed regions are necessary to validate the signal of introgression. I suggest the authors to use S^* used in human, and rIBD described in Bosse et al 2014. Many homologous regions that are challenge for variant calling will be commonly presented, to actually false positive.

We thank the reviewer for suggesting additional analyses that could strengthen our results. We agree that there are many more methods available to validate gene flow signals, but many of them will likely generate the same results because the underlying assumptions in these programs are very similar. However, we do acknowledge that a few additional analyses are

appropriate to ensure consistence of the results. In the revised manuscript, we therefore included the following additional analysis: fd-statstics together with our previous sliding-window D-statstics (Sup Figure 6,10-15), Treemix (Sup Figure 5) along with our previous G-Phocs analysis to explore models of species divergence and admixture that might accommodate the Suidae evolutionary history. We also applied a more comprehensive method described in Ní Leathlobhair et al. 2018 to test all possible historical demography scenario. These additional analyses are all very consistent with our original results, and thus strengthen the solid support of the observed introgression signals (line 185-197, Sup Figure 20-22).

Given our low coverage genomic dataset (average ~10X) of some of the samples and the relatively small sample size it will not be possible to reconstruct reliable phased genomes. Thus, although haplotype-based methods, like RFMIX, S* and rIBD as suggested by the reviewer could result in a finer inference of introgressed regions, they are unfortunately infeasible for our dataset.

4) Is there any reference reporting that pygmy hog can hybrid with pig/wild boar?

So far, there is no report about current hybridization between pygmy hog and wild boar.

5) Excluding the possibility of incomplete lineage sorting as described in Huerta-Sanchez et al (2014, nature) is necessary to infer admixture.

We thank the reviewer for this very constructive suggestion. Based on the equation described in Huerta-Sánchez et al (2014, nature), we calculated the maximum length of a haplotype with a probability < 0.005 , which could be shared by pygmy hog and wild boar due to incomplete lineage sorting (ILS). Because of the deep divergence split between pygmy hog and wild boar (at least 4.2My), the probability of a haplotype sharing with a length > 688 bp as a result of ILS is extremely small. In our sliding-window D-statistic analysis, we used a window size as 100 kb, which is much longer than the estimated ILS length (688bp). In our Saguario analysis, we furthermore filtered out all the segment having an alternative topology based on segment length ≤ 688 bp. We have included these calculations in Materials and Methods section (line 211-219).

We also follow the other approach to assess the probability of ILS as described in the Huerta-Sánchez's study. The method consists of comparing D-statistics between populations under simulations of a demographic model. One assumption in this approach is the requirement of a very detailed and precise demographic model to obtain the best assessment. The historical demographic information of pygmy hog and the ancestral population of Suinae species are still deficient. Therefore, we can only fit in a simplified model. Furthermore, inaccuracy of the effective ancestral population size and bottleneck event may lead to over-/underestimation of ILS. With this in mind, we simulated 10,000 loci with length of 100 kb under the mode described in Supplementary Note, and calculated D-statistics with the same quadruplets we used in sliding-window analysis. All simulations results resulted in $P < 0.001$ against ILS for all comparisons.

We therefore are confident that we have excluded ILS in our results which strengthens the authenticity of the gene-flow we observed.

- 6) **The authors sequenced the genomes of Duroc, Large white, Meishan, Xiang, but it is unclear why the authors chose them for sequencing. Using one individual of Duroc, Large white, Meishan, Xiang, to represent European, northern Chinese and southern Chinese pigs is dangerous. There are many released high coverage genomes of domestic pig and wild boar across Europe and China, e.g. Ai et al 2014 (Nat genet), Leno-Colorado et al (G3 2017 7(7):2171-2184). Why not use these for analysis? In addition, I am afraid that Meishan is not a Northern Chinese breed. The most famous Northern Chinese breed is Min.**

We apologize for the unclear description of our taxon sampling. In our study, we sequenced six pygmy hogs and one Babyrousa Babyrousa. The remaining samples have previously been published by our research group. We chose these published data because we are more aware of the evolutionary history of these samples, especially for the potential gene flow events (Frantz LA et al, Genome biology. 2013 Sep;14(9):R107.). With the prior knowledge, we could better fine-tune the parameters we used in the current study. In addition, we have 8 European wild boars, 2 European domesticated pigs, 2 Northern China wild boars, 1 Meishan representing Northern China domesticated pig, 2 Southern China wild boars and 1 Xiang representing Southern China domesticated pig in our dataset. Thus, we have a dataset that covers most of the geographical regions and pig breeds, but keeps the number of samples workable for the intensive analysis that we performed.

The reason for using Meishan to represent Northern China domesticated pig, is that Meishan has been reported to have a truncated haplotype on the X chromosome (Ai et al 2014, Nat genet). We included Meishan to further investigate this phenomenon. We agree that Min is the most famous Northeastern China domesticated breed. However, Min pigs have clear genomic signatures of admixture with European pigs that are genetically divergent from Chinese pigs (Yang et al. 2011; Ai et al. 2013). Based on the breeding history, Min pigs used to cross with Laiwu (typical central China domesticated pig) and recently with European pigs. Thus, in terms of genetic background, Meishan is more homogeneous compared to Min pigs.

- 7) **line 13-140. Description in ref 42:" we found an exceptionally large (14-Mb) region with a low recombination rate on the X chromosome that appears to have two distinct haplotypes in the high- and low-latitude populations, possibly underlying their adaptation to cold and hot environments, respectively." No South Asia pig was studied in this paper.**

We do not understand this comment. In ref 42 (Ai et al 2014, Nat genet) the authors compared European, Northern China and Southern China pigs. In our study, we also included Southern China wild boars/domesticated pigs to investigate potential gene flow events. Thus, we do not understand what is meant with "No South Asia pig was studied in this paper".

- 8) **Section "Wild boar harbors genetic introgression from a ghost lineage." The finding of X chr introgression has been described in Ai et al (2014). I don't think there is any important discovery here.**

We agree that the genealogical discordance in the X chromosome has been described in Ai et al (2014), and they also raised the hypothesis that the European/northern haplotype was likely introgressed from an extinct Sus suid. We also consider this paper as a very important prior

*work for our own study and cited this paper at the very beginning of the section including X chromosome introgression. With genome sequences of pygmy hogs and *Babirusa babyrussa*, we are able to further confirm Ai et al's hypothesis and more accurately locate the phylogenetic placement of the introgression donor. Moreover, the introgression signal we observed in autosomes emphasizes the ubiquity of inter-species hybridization during speciation. Thus, we think our manuscript has its own scientific value and has its own guiding significance for future studies.*

9) Figure 3b, why using unrooted tree? It is difficult to understand the relationship using unrooted tree.

We agree with the reviewer that the unrooted trees in this figure are difficult to interpreted. In the revised manuscript, we have now included rooted phylogenies using warthog as the outgroup.

Sincerely,

Langqing Liu and Ole Madsen

REVIEWERS' COMMENTS:

Reviewer #1 (Remarks to the Author):

The authors have addressed my major concerns adequately and I believe that the manuscript has been greatly strengthened as a result of these efforts.

Last but not least, although the author's argumentation is partially convincing, I still believe that an effective title should use descriptive terms and phrases that accurately highlight the key concepts of the paper, and reflect the tone of the research article and of the scholarly journal. I think the "a big piece in resolving the complex puzzle" is overly clever and less-formal, which maybe more suitable for the editorial or opinion piece.

In short, I'm happy to recommend it for publication in Nature Communications.

Mingzhou Li

Reviewer #2 (Remarks to the Author):

The authors have conveniently addressed the comments when revising the paper. However, as they do not have any information about the absence of crossbreeding in the ancestry of zoo samples, I would recommend to briefly discuss the possible impact of such event on their result.

Reviewer #3 (Remarks to the Author):

I have carefully examined the revised manuscript and your response to reviewers. It turns out that my concerns have been well addressed. So I have no further questions. I recommend this manuscript to be accepted.

Dear Editor and Reviewers:

We would like to thank the three reviewers for their positive and constructive comments and suggestions which have been very helpful in revising and improving our manuscript. Following are the point-by-point responses to reviewers' comments.

Reviewer #1:

- 1. The authors have addressed my major concerns adequately and I believe that the manuscript has been greatly strengthened as a result of these efforts.**

Last but not least, although the author's argumentation is partially convincing, I still believe that an effective title should use descriptive terms and phrases that accurately highlight the key concepts of the paper, and reflect the tone of the research article and of the scholarly journal. I think the "a big piece in resolving the complex puzzle" is overly clever and less-formal, which maybe more suitable for the editorial or opinion piece.

In short, I'm happy to recommend it for publication in Nature Communications.

Mingzhou Li

*Many thanks for your perspicacious scientific insight. Your efforts are greatly appreciated. We accept that the title should be more scientific informative. In the revised manuscript, we changed the title to "Genomic analysis on pygmy hog (*Porcula salvania*) reveals multiple interbreeding during wild boar expansion".*

Reviewer #2:

- 1. The authors have conveniently addressed the comments when revising the paper. However, as they do not have any information about the absence of crossbreeding in the ancestry of zoo samples, I would recommend to briefly discuss the possible impact of such event on their result.**

*We again thank the reviewer for reviewing the manuscript. We acknowledge that we do not have direct evidence that the sample is not a hybrid individual. However, from other analysis on genetic diversity and inbreeding (e.g. regions of homozygosity) we did not find any indication of potential genealogical discordance caused by recent crossbreeding, as the genomic diversity landscape of *Babyrussa* sample is very homogeneous, which means that the distance between both copies of the genome is consistent over the full genome (Mirte Bosses et al, unpublished results). At the same time, we did not include *Babyrussa* when detecting gene-flow signal, for example D-statistic and Saguaro. In the revised manuscript, we added corresponding explanation on this in Methods section. (line 314-316)*

Reviewer #3:

- 1. I have carefully examined the revised manuscript and your response to reviewers. It turns out that my concerns have been well addressed. So I have no further questions. I recommend this manuscript to be accepted.**

We thank the Reviewer for reviewing the revised manuscript. We believe the manuscript has really benefited from his/her comments, which were really constructive!

Sincerely,

Langqing Liu and Ole Madsen